# Intergenerational Associations between Parents’ and Children’s Adverse Childhood Experience Scores

**DOI:** 10.3390/children8090747

**Published:** 2021-08-29

**Authors:** Adam Schickedanz, José J. Escarce, Neal Halfon, Narayan Sastry, Paul J. Chung

**Affiliations:** 1Department of Pediatrics, David Geffen School of Medicine at UCLA, Los Angeles, CA 90024, USA; nhalfon@ucla.edu (N.H.); paul.j.chung@kp.org (P.J.C.); 2Department of Internal Medicine, David Geffen School of Medicine at UCLA, Los Angeles, CA 90024, USA; jescarce@mednet.ucla.edu; 3RAND Corporation, Santa Monica, CA 90401, USA; 4Department of Health Policy & Management, UCLA Fielding School of Public Health, Los Angeles, CA 90024, USA; 5Institute for Social Research, University of Michigan, Ann Arbor, MI 48104, USA; nsastry@umich.edu; 6Kaiser Permanente School of Medicine, Pasadena, CA 91101, USA

**Keywords:** adverse childhood experiences (ACEs), intergenerational, adversity, parenting, trauma, mental health

## Abstract

Background: Adverse childhood experiences (ACEs) are stressful childhood events associated with behavioral, mental, and physical illness. Parent experiences of adversity may indicate a child’s adversity risk, but little evidence exists on intergenerational links between parents’ and children’s ACEs. This study examines these intergenerational ACE associations, as well as parent factors that mediate them. Methods: The Panel Study of Income Dynamics (PSID) 2013 Main Interview and the linked PSID Childhood Retrospective Circumstances Study collected parent and child ACE information. Parent scores on the Aggravation in Parenting Scale, Parent Disagreement Scale, and the Kessler-6 Scale of Emotional Distress were linked through the PSID 1997, 2002, and 2014 PSID Childhood Development Supplements. Multivariate linear and multinomial logistic regression models estimated adjusted associations between parent and child ACE scores. Results: Among 2205 parent-child dyads, children of parents with four or more ACEs had 3.25-fold (23.1% [95% CI 15.9–30.4] versus 7.1% [4.4–9.8], *p*-value 0.001) higher risk of experiencing four or more ACEs themselves, compared to children of parents without ACEs. Parent aggravation, disagreement, and emotional distress were partial mediators. Conclusions: Parents with higher ACE scores are far more likely to have children with higher ACEs. Addressing parenting stress, aggravation, and discord may interrupt intergenerational adversity cycles.

## 1. Introduction

Adverse childhood experiences (ACEs) are stressful and potentially traumatic events, including abuse, neglect, and exposure to household dysfunction, that occur any time before age eighteen. Adverse childhood experiences are associated with higher risk of worse mental and physical health problems in adulthood and have been shown to predict a number of significant adverse outcomes over the lifecourse, including greater risk-taking behavior, worse mental health, riskier health related behaviors, greater chronic disease burden, and premature mortality [1,2]. In addition to conferring health risk upon individuals who experience adversity that ACEs measure, there is also evidence to suggest that the experience of adversity in childhood can result in a higher likelihood of perpetuating cycles of adversity for one’s children when in a parenting role. Certain ACEs may be associated with parenting practices across generations, the potential for child abuse and neglect of parents’ own children [3], and those children’s mental health and substance abuse [4]. While studies have focused on intergenerational associations for a few specific ACEs, the aggregation of different types of adverse childhood experiences into the ACE score provides a more comprehensive tool to assess risk for cross-generational transmission of adversity from parents to children. There is also rationale for considering the full ACE score, not just individual ACEs, when measuring the sum of adversity an individual’s experiences because various types of adversity are thought to impact health hazards through overlapping risk pathways, such as the final common pathways of the endocrine stress response and increased allostatic load [5,6].

No published study has measured the intergenerational associations between overall ACE scores in parents and their children in adulthood. If it were shown to be linked across generations, the well-established ACE score measure for parents could be used as an early indicator of their children’s risk of adversity, maltreatment, and household dysfunction. Unlike many screening and intervention approaches for risk of child maltreatment that are implemented in pediatric practices [7,8,9], parent ACE score is measurable even before birth and could be implemented in the prenatal setting or earlier to help target interventions to reduce the risk of intergenerational transmission of adversity.

Parental history of maltreatment in childhood has been shown to correlate strongly with parenting behaviors and risk of intergenerational transmission of child maltreatment [10,11]. Parenting frustration, anger, and psychological distress have all been shown to function as mediators that increase the risk that parents will display adverse parenting behaviors [12,13]. These associations suggest that maltreatment and other experiences of adversity lead to parenting behaviors that have the potential to perpetuate maltreatment and adversity across generations, perhaps modeled on exposures to adverse parenting and facilitated by psychological responses to those exposures. To date there has been no examination of this full cascade of intergenerational ACE transmission including its mediators.

In our study, we examine the association between parents’ ACE scores and their adult children’s ACE scores in a national sample of families, as well as potential mediators of these associations including parental mental health, parenting aggravation, and parent disagreement.

## 2. Materials and Methods

### 2.1. Design and Participants

We used data from the Panel Study of Income Dynamics (PSID), a nationally-representative panel survey with genealogic design with a sample of U.S. families beginning in 1968. Data collected by phone from the 2013 PSID main interview included information on health, education, income, health insurance, family structure, and demographic characteristics for adult heads of household, their spouses, and their children or other cohabitants. A total of 12,985 individuals who were age 19 or older and English-speaking heads of household or their spouses in the 2013 PSID main interview were eligible for the 2014 Childhood Retrospective Circumstances Study (CRCS), which retrospectively assessed childhood experiences, including nine ACEs. A total of 8072 individuals completed CRCS via web-based or mailed paper questionnaire between May, 2014, and January, 2015, for an unweighted response rate of 62 percent (weighted response rate 67%) similar to response rates for web-based supplements to other national panel studies (PSID, 2017; [14]).

Among the 8072 PSID CRCS participants whose ACEs information was collected, 2205 (27%) had a mother, father, or both who were also CRCS participants. These parent-child dyads formed the primary analytic sample for this study.

Information on parent mental health (Kessler-6 emotional distress scale), Aggravation in Parenting scores, and Parent Disagreement scores were obtained from PSID’s 1997, 2002, and 2014 waves of the PSID’s Child Development Supplement (CDS) for parents participating in the CRCS. The 1997 and 2002 waves of the CDS collected information from a single cohort of children from PSID families, and the 2014 wave followed an entirely new cohort. All waves of the CDS employed phone and in-person interviews to collect information on children’s behavior, psychological and social well-being, health status, family environment, education, and caregiver characteristics.

To examine mediators of intergenerational parent-child ACE correlations, we examined samples that included parent-child dyads in which parent mood and behavior data were collected through any wave of CDS. Of children participating in the CDS-2014, 2466 (63%) had a mother, father, or both who participated in CRCS. This sample allowed us to examine associations between parents’ ACEs and parent mood, aggravation, and disagreement during childrearing. Of the 2205 parent-child dyads with ACE information in CRCS, 660 (30%) of those dyads included children who participated in the original CDS cohort. This sub-sample allowed us to examine the degree to which parent mood, aggravation, and disagreement mediated the parent-child ACE association.

### 2.2. Construction of Adverse Childhood Experience Variable

Complete conventional ACE information in the CRCS was collected from adults who reported experiences prior to age eighteen including physical abuse, emotional abuse, sexual abuse or assault, emotional neglect, witnessing intimate partner violence in the home, witnessing household substance use, having a parent with mental illness, any parental separation or divorce, and having a deceased parent or a parent they never knew (Table 1). Adverse childhood experience counts were binned into four categories—zero, 1, 2 or 3, and 4 or more ACEs — for both the parent ACE predictor and child ACE outcome in the main analysis, similar to prior studies [15,16]. In secondary analyses, we also examined the ordinal ACE count with categories of 1 ACE through 9 ACEs for parents or children.

The parent ACEs predictor variable was specified primarily as the highest of either parent’s ACE score category, allowing for those children with only one parent in the household or only one parent who responded to CRCS to be included in the main analyses. For analyses examining the relationship between each parent’s ACE score separately with child ACEs, we included the ACE score of each parent (if present) in the model along with an indicator variable for the presence of each parent.

Within the full CRCS sample, using logistic regression models adjusted for covariates described in our main study below, we validated relationships between ACEs and chronic medical conditions (Appendix A).

### 2.3. Covariates

The PSID main interview and CDS collected covariates for our analytic models, including a five-category education variable for each parent; a continuous child years of age variable; a four-category child race variable; a binary indicator for child Latino/Hispanic ethnicity; a five-category household income variable; and count variables for number of household members and children.

Four hundred fifty-five (21%) of parent-child dyads in CRCS were missing primary predictor or outcome data. We found no significant differences in covariate composition of the sample with or without these individuals, so they were excluded. In Table 2, we present demographic differences between the sub-sample of children in parent-child dyads (2205) and those in CRCS parent-child dyads with mediator data captured in CDS (660).

### 2.4. Statistical Analyses

For our main analyses, we regressed children’s binned ACE scores on their parents’ ACE score categories using multvariable multinomial logistic regression models, adjusted for covariates and calculating survey-robust standard errors. Absolute risk and relative risk of child ACE counts by parent ACE counts were calculated via the delta method postestimation. We used the CRCS sampling weights to accommodate the complex survey design, achieve population representation, and adjust for nonresponse. We used similar multinomial logistic regression models to examine the relationship between mothers’ ACE scores and fathers’ ACE scores separately as predictor variables for the child ACE outcome.

Secondary analyses examined linear relationships between parents’ ordinal ACE counts (with categories of 1 ACE through 9 ACEs) and their children’s ordinal ACE counts and included a term for the interaction of the mother’s ACE count by the father’s ACE count and indicators of the presence of both parents.

We performed mediation analyses to assess whether parents’ mental illness symptom scores on the Kessler-6 twenty-four-point scale of emotional distress [17], their scores on the Aggravation in Parenting Scale (APS), or their scores on the Parental Disagreement Scale (PDS) mediated any relationships between parent ACE scores and behavioral health outcomes. The Kessler-6 (K6) measures psychological distress, particularly anxiety and depression symptoms, based on responses to six items each scored on a 5-point Likert scale [17]. The Aggravation in Parenting Scale is a composite average of responses between 1 (not at all true) and 5 (completely true) to each of seven items based on items from the Parenting Stress Index, asking parents how much they felt the child was harder to care for than expected, did things that bother the parent, and how much the parent feels he/she is giving up much more of life to be a parent than expected, among other negative sentiments [18]. The APS has been validated in a number of studies and found to have high reliability [19,20]. The Parental Disagreement Scale consists of 13 items assessing the extent of disagreement between the primary child caregiver and her/his spouse or partner. The items were derived from the National Longitudinal Survey of Youth and combined as the average score on a five-point Likert scale Appendix B; [21]. We conducted Sobel-Goodman mediation tests to estimate the proportion of the parent ACE score effect on child ACE scores that was mediated by continuous score on each of the PDS, APS, and Kessler-6 scores. We used maternal ACE scores for these analyses. We confirmed findings of partial mediation by loading PDS, APS, and Kessler-6 score separately into our main regression models and observed parent ACE coefficient changes.

All analyses were carried out in Stata, version 14 (StataCorp, College Station, TX, USA). The UCLA Institutional Review Board approved this study using restricted data under contract from the University of Michigan’s Institute for Social Research.

## 3. Results

The study sample for our main analyses included 2205 adult children in parent-child dyads for which the child’s ACE score and one or more parent’s ACE score was captured through the PSID CRCS. A tenth of the adult children in the sample reported experiencing four or more ACEs, while approximately a third reported none. Mothers reported more ACEs than fathers. Most individuals in the sample were white, about two thirds had some education beyond a high school degree, and almost half rated their socioeconomic status while growing up as about average (Table 2). Compared to the demographics of the overall CRCS cohort, the study sample of adult children was on average slightly more educated and was ten years younger. The sub-sample of CRCS parent-child dyads that participated in CDS in 1997/2002 was more diverse racially and ethnically, better educated, and was made up of some of the youngest participants in CRCS.

Among CRCS parent-child dyads, absolute risks of child ACE count categories by the highest of either parent’s ACE count are presented in Table 3 (See Appendix C Table A1 for adjusted relative risk ratios produced by the source multinomial logit model). Increased risk of higher child ACE counts was observed when parents had ACE counts above two, with the largest shift found when parents reported four or more ACEs. Compared to children whose parents reported no ACES, those whose parents had the highest ACE counts were one third less likely to report no ACEs (adjusted absolute risk difference 15.8 percentage points [95% CI 6.9–24.6], *p*< 0.001) and 3.3 times more likely to have four or more ACEs themselves (adjusted absolute risk difference of 16 percentage points [95% CI 8.3–23.7], *p* < 0.001).

Mothers’ ACE counts were more strongly associated with their children’s ACE counts than were fathers’ ACE counts (Table 3) when included separately in the multinomial logistic model. Children whose mothers reported four or more ACEs were forty percent less likely to report no ACEs (95% CI 0.37–0.81, *p* < 0.001) and 4.76-fold more likely to report four or more ACEs themselves (95% CI 2.5–7.0, *p* < 0.001), compared to those whose mothers reported no ACEs. Paternal ACE counts showed an overall positive association with their children’s ACE counts, but the effect was less pronounced than for mothers’ ACE counts (see Appendix C Table A2 for multinomial logit model results).

Children whose parents both reported four or more ACEs had a 7.7-fold (95% CI 2.1–13.4; *p* = 0.007) and 40.5 percentage point (95% CI 15.9–65.1, *p* = 0.001) increased risk of reporting four or more ACEs themselves compared to children whose parents experienced no ACEs between them. Figure 1 plots the absolute risk of children reporting four or more ACEs by paternal ACE counts and maternal ACE counts.

In secondary linear models with ordinal child ACE counts as the outcome and interacting mothers’ ACEs by fathers’ ACEs, each additional maternal ACE was associated with an average increase of one quarter of an ACE in their children (0.25, 95% CI [0.17–0.33], *p* < 0.001), while each additional paternal ACE was associated with an average increase of one fifth of an ACE (0.19, 95% CI [0.08–0.30], *p* = 0.001). The coefficient on the mother’s-ACE-count-by-father’s-ACE-count interaction variable in the model was negative (−0.09, 95% CI [−0.14, −0.05], *p* < 0.001), indicating that there is a dampening in the unit change in the effect of one parent’s ACE count on a child’s ACE count outcome associated with an increase in the ACE count of the other parent.

Among parents of children in the 2014 CDS, parent ACE count was positively associated with scale scores on each of the three mediator variables—the Kessler-6 scale of psychological distress, the Aggravation in Parenting Scale, and the Parental Disagreement Scale. Mothers and fathers who reported four or more ACEs were found to have higher Kessler-6 scale scores than mothers and fathers who reported experiencing no ACEs. Higher paternal ACE counts were not as consistently associated with changes in Aggravation in Parenting scale scores but higher maternal ACE counts were. Maternal ACE counts showed a positive association with or Parent Disagreement Scale scores at all levels of maternal ACE counts (Table 4).

In the sub-sample of CRCS parent-child dyads that also participated in CDS almost two decades prior, formal Sobel-Goodman mediation analyses showed that 21% of the association between child ACE count and maternal ACE count was mediated by the children’s primary caregivers’ (typically their mothers) scores on the Aggravation in Parenting Scale, 31% of the association was mediated by Kessler-6 scale of emotional distress scores, and 44% of the association was mediated by scores on the Parental Disagreement Scale.

## 4. Discussion

In this study of a national sample of parent-child dyads we found that parents’ ACE counts were positively correlated with their children’s ACE counts, adjusting for demographic factors and socioeconomic status. Maternal ACE counts showed a stronger association with children’s ACE counts than paternal ACE counts, but together both parents’ ACE counts were predictive of child ACE count risk.

This is the first report quantifying the association between parents’ ACE counts and their children’s ACE counts. It extends a literature showing that certain kinds of adversity, such as physical abuse and mental health problems, are linked across generations within families. Our findings build on a large literature demonstrating the influence of childhood experiences on later parenting [22,23] and parent factors that put children at risk for early life adversity [24,25].

We confirm partial mediators of these intergenerational ACE associations including parental mental health, aggravation toward their children and with parenting, and measures of parenting conflict. Future studies should examine not only adversity risks as mediators of intergenerational ACE associations but also protective factors, such as parenting support and measures of resilience.

Maternal ACE counts were more strongly associated with child ACE counts than paternal ACE counts, which could be due to differences in parenting roles, differences in parenting behaviors, greater likelihood of mothers remaining the sole parent in single-parent households compared to fathers, or hereditary transmission of risk factors in utero [26]. The dampening of intergenerational ACE associations when both parents report higher ACE counts, suggests that two-parent households are protective against ACE transmission to children or, perhaps, that in households where both parents have experienced more childhood adversity there are other factors that reduce this association, such as greater resilience or coping despite this history of adversity.

In 2012 the American Academy of Pediatrics issued broad recommendations for a two-generation approach to identifying high risk families that encompassed asking children and parents about experiences of early adversity, which could include screening for ACEs [27]. However, only fifteen percent of pediatric practices regularly screen for more than two parent ACEs [28]. Our findings lend further evidence in support of screening for parents’ ACE scores to risk-stratify children according to their likelihood of experiencing adversity, maltreatment, and household dysfunction. Screening parents for a full set of ACEs could provide opportunities to anticipate and interrupt the intergenerational cycle of adversity that ACEs may initiate and perpetuate, as well as the downstream health consequences of childhood adversity such as greater burden of mental health issues, substance use, chronic illness, and premature mortality. If these hazards to lifelong success can be traced back, even just in part, to parent ACEs, this could help target prevention early in an at-risk child’s life, perhaps by equipping parents with parenting skills to minimize the risk of maltreatment before children are even born.

A growing evidence base has demonstrated the effectiveness of interventions designed to prevent child maltreatment in high-risk families in which parents have experienced significant early childhood adversity themselves [29], but the most effective programs must be implemented soon after birth and are resource intensive [30]. Parental ACE screening could be used to risk-stratify and focus home visitation and other program resources on children in families with the highest likelihood of ACE transmission.

Our findings further support a growing literature on family-based, two-generation approaches to mental illness treatment. Intervention during the perinatal period has been suggested as a method to reduce adult mental illness burden, and our finding of intergenerational associations between ACE scores (including measures of mental illness) suggest that focusing behavioral and mental health resources very early on in children whose parents had high ACE counts could be an effective strategy for reducing the burden of mental illness [31]. Clinically validated and implemented approaches to preventing child maltreatment and exposure to violence may represent strategies to not only minimize harms to the child in the short term but also prevent ACEs in future generations [32]. Interventions that focus on treating parent mental illness, helping parents cope with aggravation in parenting, and reducing parental discord may be especially effective for interrupting the intergenerational transmission of childhood adversity.

### Limitations

Our study has a number of limitations. Despite being the only study to our knowledge with adult child and parent ACE information from each of the conventional ACE domains, we relied on sub-samples of CRCS to estimate intergenerational parent-child ACE associations and their mediators because no complete sample exists with all the requisite data for this study. Restricting our sample to those CRCS participants who were part of parent-child dyads could have introduced selection bias. Our analysis, and the ACEs literature overall, relies on a retrospectively reported measures that could introduce recall bias. Reverse causality and unmeasured confounding are potential threats of these retrospectively reported ACEs, though the longitudinal nature of our dataset allowed us to examine correlates of childhood adversity, such as parental mental health, that correlated with retrospectively reported ACEs but were collected at the time the ACEs were occurring (i.e., during childhood/childrearing). Other background confounders, such as poverty and low educational attainment, associated with ACEs that multiple family generations may experience in common could drive the correlation between ACEs across generations but were controlled in our study. Further, poverty, racism, and other social and structural determinants of health have themselves been posited as putative ACEs due to their influence on health and the lifelong stress they introduce, but those factors were not included in our ACE score variable because they have not yet been widely adopted as conventional ACE categories. Also, the protective impact of resilience factors that buffer ACEs is well-recognized, but our study did not include measures of resilience factors in addition to ACE measures. The timing of collection of our mediator variables may not have corresponded with the timing of ACE exposures in children. We cannot identify sensitive developmental windows or time-varying risk factors that could have driven ACE counts or potentiated their impacts.

## 5. Conclusions

Our study is the first to demonstrate clear correlations between overall parent ACEs and ACE counts in their children. Mother’s ACEs were more strongly correlated with their children’s ACE counts than fathers’, but each parent ACE score showed an additive effect in increasing children’s ACE risk. Parent mental health, aggravation in parenting, and parenting disagreement each partially mediated the intergenerational ACE score correlation, suggesting that they contribute to intergenerational ACE score associations between parents and their children. Early identification of these childhood ACE risks from parental history could provide opportunities for early intervention to reduce intergenerational transmission of ACEs by focusing on improved parental mental health, reducing aggravation over parenting roles, and helping parents minimize disagreements and conflicts.

## Figures and Tables

**Figure 1 children-08-00747-f001:**
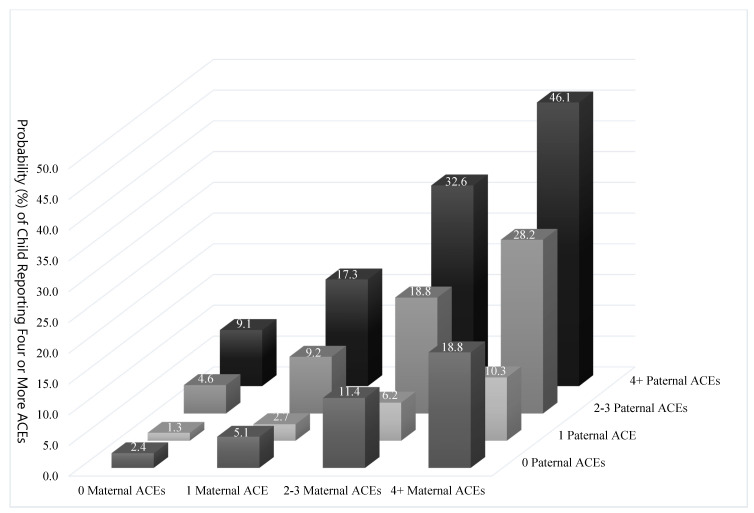
Risk of Four or More Child ACEs by Paternal and Maternal ACE Complement.

**Table 1 children-08-00747-t001:** Adverse Childhood Experiences (ACE) Categories and Rates of ACEs in Childhood Retrospectives Circumstances Study (2014–2015, *n* = 7223).

PSID CRCS	ACE Category	Description Based on ACE Survey Item	Weighted Percentage Positive
ACE Type	Emotional Abuse	Respondent rating their relationship as poor with their mother and/or father and indicating that the relationship involved the highest degree of emotional tension	3%
	Physical Abuse	Mother and/or father sometimes or often slapped, threw things at, or otherwise physically harmed the respondent	23.1%
	Sexual Abuse	Respondent reported being the victim of a crime classified as assault or rape in childhood	3.6%
	Intimate Partner Violence	Respondent reported that his/her mother and father often, sometimes, or not very often pushed, threw things at, or were otherwise physically harmful toward one another	20.8%
	Household Substance Abuse	Respondent reported his/her mother and/or father abused drugs or alcohol	19.5%
	Mental Illness in Household	Respondent reported his/her mother and/or father had any mental health problems (panic attacks, depression)	21.4%
	Parental Separation or Divorce	Respondent reported his/her parents were separated or divorced	27%
	Emotional Neglect	Respondent reported that his/her mother or father displayed no affection or parenting effort	7.2%
	Deceased or Absent Parent	Respondent reported that his/her mother or father was deceased or unknown to him/her at a time in the childhood of the respondent	5%

**Table 2 children-08-00747-t002:** Sample Characteristics for Parent-Child Dyads in Childhood Retrospective Circumstances Study.

	Weighted Percentage or Mean (Standard Deviation) for Sample of Parent-Child Dyads with ACEs Data (*n* = 2205)	Weighted Percentage or Mean (Standard Deviation) for Parent-Child Dyads with ACEs Data and Mediation Variable Data (*n* = 660)
Adult Child Characteristics		
Female	52.1	55.2
Race		
White	85.8	81.6
African American	11.8	9.6
Asian/Pacific Islander	1.6	5
Other	0.9	3.9
Latino/Hispanic	4	15.7
Adult Child’s Education		
Less Than High School	6.5	7.5
High School Graduate or Equivalent	22.9	18.8
College/Vocational School/Graduate School	70.6	73.8
Adult Child’s Age in Years (Mean (SD))	39.4 years (11.2)	25.1 years (2.3)
Number of Adverse Childhood Experiences in Adult Child		
0	36.9	39.4
1	28.5	22.8
2–3	24.1	29.5
4 or more	10.5	8.3
Mother’s Education		
Less Than High School	11.6	4
High School Graduate/GED	28.2	16.9
College/Vocational/Graduate School	60.2	79.1
Father’s Education		
Less Than High School	10.6	5.3
High School Graduate/GED	24.2	18.5
College/Vocational/Graduate School	65.3	76.2
Number of Adverse Childhood Experiences of Mother		
0	40	46.9
1	29.5	19.8
2–3	21.1	25.5
4 or more	9.4	7.8
Number of Adverse Childhood Experiences of Father		
0	41.7	35.9
1	29.4	36
2–3	22.4	24.6
4 or more	6.5	3.5

**Table 3 children-08-00747-t003:** Absolute Risk of Child ACE Count Category by Highest of Either Parent’s ACE Count Category.

Estimates of Absolute Risk (95% CI) (Column Totals Sum to One Hundred Percent)	Higher of Either Parent’s Adverse Childhood Experience Score			
	0 ACEs	1 ACE	2–3 ACEs	4 or More ACEs
Probability of 0 Child ACEs	43.8% (38.7–48.9)	41.2% (36.4–46.0)	31.6% (27.1–36.2)	28.0% (20.8–35.2)
Probability of 1 Child ACE	25.3% (20.6–29.9)	30.3% (25.5–35.1)	29.6% (24.5–34.7)	25.8% (18.1–33.6)
Probability of 2–3 Child ACEs	23.8% (19.3–28.3)	22.7% (18.5–26.9)	24.1% (19.6–28.6)	23.0% (16.2–30.0)
Probability of 4+ Child ACEs	7.1% (4.4–9.8)	5.8% (3.5–8.2)	14.6% (10.5–18.7)	23.1% (15.9–30.4)
	**Mothers’ ACE Score**			
Probability of 0 Child ACEs	42.6% (38.4–46.7)	37.8% (32.4–46.7)	30.4% (24.7–36.1)	25.1% (16.2–34.0)
Probability of 1 Child ACE	25.8% (21.9–29.6)	29.7% (23.9–35.5)	28.4% (22.2–34.7)	23.2% (14.1–32.2)
Probability of 2–3 Child ACEs	25.8% (21.9–29.7)	21.5% (16.3–26.7)	21.3% (15.7–26.8)	23.8% (15.6–32.0)
Probability of 4 or More Child ACEs	5.8% (4.0–7.7)	11.0% (6.3–15.6)	19.9% (13.7–26.1)	27.9% (18.9–36.9)
	**Fathers’** **ACE Score**			
Probability of 0 Child ACEs	39.2% (35.7–42.7)	39.4% (32.1–46.7)	31.8% (25.2–38.5)	32.4% (18.5–46.2)
Probability of 1 Child ACE	27.5% (24.1–30.9)	27.1% (19.3–34.9)	25.8% (17.9–33.8)	26.5% (11.0–42.1)
Probability of 2–3 Child ACEs	22.5% (19.3–25.6)	28.4% (20.4–36.3)	30.5% (22.1–38.8)	19.8% (5.7–33.9)
Probability of 4 or More Child ACEs	10.9% (8.5–13.2)	5.1% (1.6–8.7)	11.9% (5.4–18.3)	21.3% (5.6–37.1)

**Table 4 children-08-00747-t004:** Associations between Parents’ Adverse Childhood Experiences Reported in the Childhood Retrospective Circumstances Survey and Mediator Measures from the Same Parents Reported in the 2014 Child Development Supplement.

Parent Psychological Distress & Attitudes Mediators (Coefficients Represent the Linear Change in Mediator Scale Score for Each Parent ACE Count Increase)	Parent Adverse Childhood Experience Count			
	0 ACEs	1 ACE	2–3 ACEs	4 or More ACEs
**Higher of Either Parent’s ACE Score** **(*n* = 2558)**				
Kessler-6 Emotional Distress Scale	Ref	0.48 (0.1–0.9) *	1.03 (0.6–1.4) ***	1.51 (1.0–2.06) ***
Aggravation in Parenting Scale	Ref	−0.001 (−0.1–0.1)	0.14 (0.04–0.2) **	0.13 (0.01–0.24) *
Parent Disagreement Scale	Ref	0.13 (0.04–0.2) **	0.07 (−0.02–0.2)	0.14 (0.04–0.2) **
**Mother’s ACE Score** **(*n* = 2296)**				
Kessler-6 Emotional Distress Scale	Ref	−0.005 (−0.4–0.4)	1.10 (0.7–1.5) ***	1.34 (0.7–2.0) ***
Aggravation in Parenting Scale	Ref	−0.15 (−0.2–−0.1) **	0.15 (0.1–0.3) **	0.15 (0.04–0.3) *
Parent Disagreement Scale	Ref	0.14 (0.1–0.2) **	0.12 (0.04–0.2) **	0.17 (0.1–0.3) ***
**Father’s ACE Score** **(*n* = 1583)**				
Kessler-6 Emotional Distress Scale	Ref	0.88 (0.5–1.3) ***	0.58 (0.1–1.1) *	1.44 (0.9–2.0) ***
Aggravation in Parenting Scale	Ref	0.15 (0.04–0.26) **	0.12 (0.00–0.23) *	−0.02 (−0.2–0.1)
Parent Disagreement Scale	Ref	0.09 (−0.01–0.2)	0.07 (−0.03–0.2)	007 (−0.1–0.2)

* indicates alpha < 0.05 threshold, ** indicates alpha < 0.01 threshold, and *** indicates alpha < 0.001 threshold.

## Data Availability

Data used in this study is restricted by the University of Michigan and accessible through its restricted data enclave via an agreement with the University of Michigan’s Institute for Social Research, given the sensitive nature of the childhood adversity information it contains.

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
