# Peer review of "Intergenerational Associations between Parents’ and Children’s Adverse Childhood Experience Scores"

_children, 2021, doi:10.3390/children8090747_

Round 1

Reviewer 1 Report

This article examines the relationship between parental and child Adverse Childhood Experiences (ACEs), as well as parental factors that may mediate this relationship. The article is very well written and the methods are described in great detail. There are a few minor areas that could benefit from clarification to strengthen the manuscript:

1) Introduction: The authors state that “no published study has measured the intergenerational associations between ACE scores in parents and their children” (page 2, line 52-53). Should this instead say “parents and their children in adulthood”? The authors note this later in the discussion as well. There are studies that examine parent and child/adolescent ACEs, such as:  

  • A Nayaran et al. (2017). Intergenerational continuity of adverse childhood experiences in homeless families: Unpacking exposure to maltreatment versus family dysfunction.
  • TJ Schofield et al. (2018). Intergenerational continuity in adverse childhood experiences and rural community environments.
  • S Negriff. (2020). Expanding our understanding of intergenerational exposure to adversity.

2) Methods: I’m not certain that “top-coding” (page 3, line 208) is a term that is widely familiar. It might be good to define this, if the authors agree.

3) Discussion:

  • I would add to the limitations that only 9 ACEs were assessed in this study, which may have influenced findings. The authors mention that the study controlled for background confounders, like poverty and education. But I would suggest briefly mentioning research shows that many other types of trauma/ACEs that were not assessed can also impact health outcomes.
  • Similarly, I would suggest discussing the protective role of resilience factors and that this was not examined in this study.

4) Table 2: In the “adult child age in years” row, add what the data in parentheses represent.

Author Response

Reviewer Comments Author  Response to Reviewer Comments
This article examines the relationship between parental and child Adverse Childhood Experiences (ACEs), as well as parental factors that may mediate this relationship. The article is very well written and the methods are described in great detail. There are a few minor areas that could benefit from clarification to strengthen the manuscript: Thank you for your review and helpful suggestions. 

1) Introduction: The authors state that “no published study has measured the intergenerational associations between ACE scores in parents and their children” (page 2, line 52-53). Should this instead say “parents and their children in adulthood”? The authors note this later in the discussion as well. There are studies that examine parent and child/adolescent ACEs, such as:  

  • A Nayaran et al. (2017). Intergenerational continuity of adverse childhood experiences in homeless families: Unpacking exposure to maltreatment versus family dysfunction.
  • TJ Schofield et al. (2018). Intergenerational continuity in adverse childhood experiences and rural community environments.
  • S Negriff. (2020). Expanding our understanding of intergenerational exposure to adversity.
Thanks for this suggestion. We have made the specified change to the text on page 2, line 54. 
2) Methods: I’m not certain that “top-coding” (page 3, line 208) is a term that is widely familiar. It might be good to define this, if the authors agree. Yes, let's avoid using the jargon of "top-coding". Instead, we've described the coding of the variables in more detail without using the "top-coding" term (Page 3, line 118; Page 4, line 155). 

3) Discussion:

  • I would add to the limitations that only 9 ACEs were assessed in this study, which may have influenced findings. The authors mention that the study controlled for background confounders, like poverty and education. But I would suggest briefly mentioning research shows that many other types of trauma/ACEs that were not assessed can also impact health outcomes.
Thank you for this welcome suggestion. I have added a note to this end on page 11, lines 323-327. 
  • Similarly, I would suggest discussing the protective role of resilience factors and that this was not examined in this study.
Excellent suggestion. We have added a note to the Limitations section to this end on page 11, lines 327-329. 
4) Table 2: In the “adult child age in years” row, add what the data in parentheses represent. Thank you for pointing this out. We have made this change in the table to indicate that the parentheses contain the standard deviation. This is noted at the top of each of the data columns in this table, but it may not be as easy to find there as it would be if listed in the row for that data (which is where I've now added that note). 

Reviewer 2 Report

This paper is very well done.  My only issue is that it would be useful to differentiate between same-sex couples and heterosexual couples at the parental level, with particular concern for the ACE of sexual abuse as an issue.

Author Response

Many thanks to the reviewer for his/her review and input on our manuscript. We are not sure we understand the suggestion provided. As best we can tell, one way to interpret the suggestion is that the reviewer would like us to re-analyze the parent dyad data to create an indicator variable of same-sex parent dyads and look for associations between that indicator variable and the individual ACE category of sexual abuse in the parents and children. This would represent a substantial departure beyond the aims of the study and would involve a completely new set of analyses looking at specific ACE categories across generations of participants, which strikes us as outside of the scope of our study. We also are not familiar with the evidence base linking same-sex parenting and sexual abuse by those same-sex parents. Evidence suggests that lifetime history of sexual abuse is more common among sexually marginalized groups (Friedman, M. S., Marshal, M. P., Guadamuz, T. E., Wei, C., Wong, C. F., Saewyc, E., & Stall, R. (2011). A meta-analysis of disparities in childhood sexual abuse, parental physical abuse, and peer victimization among sexual minority and sexual nonminority individuals. American journal of public health, 101(8), 1481–1494. https://doi.org/10.2105/AJPH.2009.190009), but to our mind this does not have clear bearing on intergenerational ACE associations. Further, our dataset is far under-powered to detect any associations between same-sex parent dyads (a small fraction of our moderate sample) and sexual abuse (one of the least common ACEs in our sample, experienced by fewer than 4% of respondents). For these reasons, we have respectfully opted to not make changes to the manuscript in response to the reviewer's suggestion. 
